# Aberrant Expression of Intracellular let-7e, miR-146a, and miR-155 Correlates with Severity of Depression in Patients with Major Depressive Disorder and Is Ameliorated after Antidepressant Treatment

**DOI:** 10.3390/cells8070647

**Published:** 2019-06-27

**Authors:** Yi-Yung Hung, Ming-Kung Wu, Meng-Chang Tsai, Ya-Ling Huang, Hong-Yo Kang

**Affiliations:** 1Department of Psychiatry, Kaohsiung Chang Gung Memorial Hospital, and Chang Gung University College of Medicine, Kaohsiung 833, Taiwan; 2Graduate Institute of Clinical Medical Sciences, College of Medicine, Chang Gung University, Kaohsiung 833, Taiwan; 3Department of Obstetrics and Gynecology, Kaohsiung Chang Gung Memorial Hospital, Kaohsiung 833, Taiwan

**Keywords:** major depressive disorder, microRNA, miR-146a, miR-155, let-7e, Toll-like receptor 4

## Abstract

Chronic inflammation and abnormalities in Toll-like receptor (TLR) signaling pathways are associated with major depressive disorder (MDD). Our previous work reported that impaired negative regulators for the TLR pathways are associated with MDD. This study aimed to assess the association between the severity of depression and the intracellular microRNAs that regulate TLR4 signaling in both peripheral blood mononuclear cells (PBMCs) and monocytes from MDD patients. The severity of MDD before and after antidepressant treatment was determined by the 17-item Hamilton Depression Rating Scale, and quantitative RT-PCR was used to measure the levels of intracellular regulatory microRNAs, including let-7e, miR-21-5p miR-145, miR-223, miR-146a, and miR-155, in PBMCs and monocytes isolated from 43 healthy controls and 84 patients with MDD before and after treatment with antidepressants. Assays of PBMCs showed that the levels of let-7e, miR-146a, and miR-155 were lower in MDD patients than in healthy controls and were significantly higher after than before treatment in the 69 patients who completed treatment with antidepressants for four weeks. Levels of miR-146a and miR-155 in monocytes were lower in MDD patients than in controls and were increased in the former after antidepressant treatment. Multiple linear regression analyses found that let-7e and miR-146a expression before treatment was inversely correlated with severity of depression, whereas miR-155 before treatment was directly correlated with severity of depression. These findings suggest that intracellular regulatory microRNAs which regulate TLR4 signaling are aberrantly expressed in patients with MDD and that these levels are ameliorated by antidepressant treatment.

## 1. Introduction

Major depressive disorder (MDD) is an important public health challenge [1]. This condition affects 4.4–20% of the general population [2], including 17% of the population of the United States [3]. The depression was found to be related to inflammation [4]. Higher levels of interleukin (IL)-1β, IL-6, tumor necrosis factor (TNF)-α, and C-Reactive Protein (CRP) were reported in patients with depression by many investigators [5,6,7,8]. Depression was also linked to the peripheral and central innate immune systems [9]. In the central nervous system, significantly increased protein expressions of Toll-like receptor (TLR) 2, TLR3, TLR4, TLR6, and TLR10, and messenger RNA (mRNA) expression of TLR2 and TLR3 were reported recently [10]. In peripheral blood, we found that mRNA levels of TLR3, TLR4, TLR5, and TLR7 were significantly increased in depressed patients [11].

TLRs, which play a critical role in early innate immunity, have important roles in the recognition of pathogen-associated molecular patterns from infectious and endogenous pathogens. Recognition of these pathogens induces the production of large amounts of various inflammatory cytokines, including IL-6, TNF-α, and IL-1β, through the activation of transcription factors, especially nuclear factor kappa B (NF-κB) [12]. The balance of TLR4 signaling pathways is strictly and finely regulated by many factors [13,14], including microRNAs.

Dysregulation of negative feedback for TLR4 signaling may exacerbate inflammation, inducing inflammatory diseases such as systemic lupus erythematosus (SLE) and rheumatoid arthritis (RA), both of which are associated with downregulation of tumor necrosis factor-α-induced protein 3 (TNFAIP3) and suppressor of cytokine signaling (SOCS) [15,16,17]. TNFAIP3 was recently shown to be an important biomarker in MDD and to be altered by treatment with antidepressants [18]. MicroRNAs negatively targeting TLR4 signaling are also associated with inflammation and several diseases. For example, miR-21-5p, a negative regulator of TLR4 via targeting of the proinflammatory tumor suppressor programmed cell death 4 (PDCD4), is thought to be important following an experimental model of autoimmune encephalomyelitis [19], and miR-146 and miR-125, which are involved in the regulation of innate immunity and inflammation, are associated with RA [20]. MiR-146a from whole blood was also reported to be associated with responses to antidepressants [21,22]. Leukocyte microRNAs that negatively regulate TLR4 signaling may also be important in MDD.

MicroRNAs play an important role in psychiatric disorders, especially depression [23,24,25,26]. Samples from peripheral blood including PBMCs, whole blood, serum, and plasma were screened to identify any microRNAs associated with depression. However, only plasma levels of the microRNAs miR-34b–5p, miR-34c–5p, miR-107, and miR-381 [27] and cerebrospinal fluid (CSF) levels of miR-16 [28] were reported to be associated with the clinical severity of depression. Some microRNAs including miR-155, miR-124, miR-451a, miR-34a-5p, and miR-221-3p serve as indicators of disease progression or therapeutic efficacy in depression [29,30,31,32]. MicroRNAs were shown to affect inflammation at different stages of signal transduction in the TLR4 pathway [33]. Let-7e and miR-223 directly negatively regulate TLR4 expression [34,35]. The targets of miR-146 are interleukin-1 receptor-associated kinase 1(IRAK1), interleukin-1 receptor-associated kinase 2 (IRAK2), tumor necrosis factor receptor associated factor 6 (TRAF6), and interferon regulatory factor 5 (IRF5) [35,36]. MiR-155 and miR-21 showed a synergistic effect on inhibiting TLR4 signaling via targeting Src homology 2 domain-containing inositol-5′-phosphatase 1 (SHIP-1) [37]. Mir-145 inhibits Toll–interleukin-1 receptor domain-containing adaptor protein (TIRAP) to reduce inflammation [38]. Although microRNA-associated pathways were analyzed in patients with depression [21], to our knowledge, limited studies assessed the effects of intracellular microRNA on TLR4 signaling related to MDD.

The enhanced secretion of proinflammatory cytokines in MDD requires both activation of TLR-mediated signaling and impaired counterbalance of negative regulators for TLR signaling. To test the hypothesis, we previously examined negative regulators of TLR4 signaling and found TNFAIP3 as an important biomarker in MDD patients [18]. Here, we aimed to investigate differences in microRNA expression profiles for negatively regulating TLR4 signaling including let-7e, miR-21-5p, miR-223, miR-145, miR-146a, and miR-155 in PBMCs and monocytes, which showed altered patterns of response to endotoxin challenge in depressed patients [10] from patients with MDD before and after treatment with antidepressants.

## 2. Materials and Methods

### 2.1. Experimental Design

Inpatients hospitalized in the psychiatric ward of Kaohsiung Chang Gung Memorial Hospital, Taiwan, with major depression from August 2013 to August 2018 were enrolled. Patients and healthy controls provided written informed consent, and the study protocol was approved by the Institutional Review Board and hospital ethics committee of Kaohsiung Chang Gung Memorial Hospital (101-5012A3, 103-5114B, and 103-6984A3; 201700153B0 and 201700539A3). Blood samples for microRNA analysis were obtained from MDD patients before and after treatment with antidepressants. Single blood samples were obtained from healthy control subjects.

### 2.2. Participants

Participants ranged in age from 20–65 years, and were medically healthy based on their clinical history, physical examination, negative routine blood tests, and urine examination. Patients with MDD were screened and enrolled by two psychiatrists before entering the study, as described previously [18,39]. Briefly, screening included a structured clinical interview for the Diagnostic and Statistical Manual of Mental Disorders, Fifth Edition (DSM-V) Axis I Disorders, an assessment of current psychiatric symptoms, and a determination of previous antidepressant treatment. The severity of MDD before and after antidepressant treatment was assessed by the same psychiatrists using the 17-item Hamilton Depression Rating Scale (HAMD-17) [40]. Remission was defined as total HAMD-17 score ≤ 7 [41]. Patients with limited capacity and psychotic disorder, mental retardation, substance dependence, severe metabolic syndrome, a body mass index (BMI) >34 kg/m^2^, a history of any systemic inflammatory disease, or who were taking anti-inflammatory or immune-modulating drugs were excluded from the study. No antidepressant taken for at least one week was reported by the patient before they entered the study. Healthy control subjects were screened by psychiatrists using the criteria of the Diagnostic and Statistical Manual of Mental Disorders (Fifth Edition) and history taking to rule out a personal or family history (first-degree relative) of psychiatric disorder.

Blood samples were obtained from patients at baseline and after foir weeks of antidepressant treatment. Patients were hospitalized in the psychiatric ward of Kaohsiung Chang Gung Memorial Hospital and monitored for drug adherence, strict regular sleep–wake cycles, a well-controlled diet, and limited smoking.

### 2.3. Treatment

Treatment was administered based on clinical considerations, in that the choice of antidepressant was not influenced by the study. Chosen antidepressants were administered and recorded after screening at baseline. The antidepressants included escitalopram (10–20 mg/day; *n* = 9), fluoxetine (40–80 mg/day; *n* = 5), paroxetine (20–40 mg/day; *n* = 6), sertraline (75 mg/day; *n* = 1) duloxetine (60–120 mg/day; *n* = 28), venlafaxine (37.5–225 mg/day; *n* = 5), bupropion (300 mg/day; *n* = 1), mirtazapine (30–45 mg/day; *n* = 4), and agomelatine (25–50 mg/day; *n* = 10). In the non-remission group, patients received escitalopram (10–20 mg/day; *n* = 5), fluoxetine (40–80 mg/day; *n* = 4), paroxetine (20–40 mg/day; *n* = 2), duloxetine (60–120 mg/day; *n* = 17), venlafaxine (37.5–225 mg/day; *n* = 3), bupropion (300 mg/day; *n* = 1), mirtazapine (45 mg/day; *n* = 1), and agomelatine (25–50 mg/day; *n* = 5). In the remission group, patients received escitalopram (10–20 mg/day; *n* = 4), fluoxetine (40–80 mg/day; *n* = 1), paroxetine (20–40 mg/day; *n* = 4), sertraline (75 mg/day; *n* = 1), duloxetine (60–120 mg/day; *n* = 11), venlafaxine (37.5–225 mg/day; *n* = 2), mirtazapine (45 mg/day; *n* = 3), and agomelatine (25–50 mg/day; *n* = 5). All benzodiazepines administered as anxiolytics or hypnotics were limited. Patients were provided one or two supportive psychotherapy sessions during the course of the study, and urged to participate in regular activities while hospitalized.

### 2.4. Quantitative Reverse-Transcription Polymerase Chain Reaction (qRT-PCR)

Venous blood (15 mL) samples were drawn between 6:00 a.m. and 10:00 a.m., after subjects fasted for 9 h. PBMCs were isolated from venous blood samples by Ficoll–Paque (GE, #17-5442-02) density gradient centrifugation and labeled with BD IMag anti-human CD14 Magnetic Particles DM (BD Biosciences, #557769) according to the manufacturer’s directions. Labeled monocytes were collected for further analysis. All samples were stored at −80 °C until assayed. RNA samples were extracted using Quick-RNA MiniPrep kits (Zymo Research R1055) and reverse-transcribed to complementary DNA (cDNA) using TaqMan Micro RNA Transcription Kits (Applied Biosystems 4366596). Specific sequences were amplified in an Applied Biosystems 7500 system using TaqMan^®^ Universal Master Mix II, without UNG (Applied Biosystems 4440040). The amplification conditions included an initial denaturation at 95 °C for 10 min, followed by 40 cycles of denaturation at 95 °C for 15 s, and annealing and extension at 6 °C for 1 min. Gene expression was quantified using the software provided by the manufacturer (Applied Biosystems), analyzed according to the △△Ct method, and normalized to the expression of U6 small nuclear RNA (snRNA). Primers for qRT-PCR are showed in Appendix A (Appendix A).

### 2.5. Statistical Analysis

All results are presented as means ± standard deviation (SD). Statistical analyses were performed using Statistical Product and Service Solutions (SPSS) version 22. Age and body mass index (BMI) were compared using Student’s *t*-tests, and sex was compared using chi-squared tests. Between-group differences were assessed by analysis of covariance (ANCOVA) following adjustment for age, sex, smoking, and BMI, and the F-value in results is defined as between-group variance divided by within-group variance. The effects of antidepressants on microRNA levels before and after treatment were tested by paired *t*-tests. A linear regression model was used to establish the relationship between baseline levels of microRNA in monocyte and baseline HAMD-17 scores; *p*-values <0.05 were regarded as statistically significant.

## 3. Results

### 3.1. Demographic and Clinical Characteristics

A total of 84 patients with MDD were enrolled, including 20 men and 64 women. Of these, 69 patients were treated with an antidepressant for four weeks and returned for follow-up examination. Age was slightly lower in the control group (41.88 ± 9.03 years) than in MDD patients before (45.20 ± 11.00 years) and after (45.56 ± 10.46 years) treatment ((1) vs. (3), (2) vs. (3), Table 1). BMI was similar in healthy controls and MDD patients before treatment. The rate of smoking was significantly higher in the MDD group. HAMD-17 scores were significantly lower after (8.91 ± 5.08) than before (24.16 ± 5.48) treatment. When patients were divided into remission and non-remission groups, there was no difference in age, sex, BMI, smoking, and HAMD score before treatment ((4) vs. (5), Table 1).

### 3.2. Levels of microRNAs and Effect of Antidepressants in PBMCs

To explore the expression profiles of intracellular microRNA regulators of TLR4 signaling, PBMCs were isolated from MDD patients and healthy controls, and their levels of microRNAs, including let-7e, miR-21-5p, miR-145, miR-223, miR-146a, and miR-155, were determined. The levels of let-7e, miR-21-5p, miR-146a, and miR-155 were significantly lower in PBMCs isolated from MDD patients at baseline than from healthy controls after adjustment for age, sex, smoking, and BMI ((1) vs. (3), Table 2). The expression of IL-6 mRNA was higher in MDD patients than in health controls. The levels of miR-145 and miR-223, however, did not differ significantly in these two groups.

To evaluate the association between antidepressant treatment and microRNA regulators of TLR4 signaling, RNA was isolated from PBMCs obtained from the 69 patients who completed the four weeks of treatment with antidepressants. Compared with baseline, the levels of let-7e, miR-223, miR-146a, and miR-155 in PBMCs were significantly increased after four weeks of therapy, and IL-6 was significantly downregulated ((1) vs. (2), Table 2).

### 3.3. Levels of microRNAs and Effects of Antidepressants in Monocytes

The expression profiles of intracellular microRNA regulators were also assessed in monocyte preparations isolated from PBMCs. Similar to the results obtained from PBMCs, miR-146a and miR-155 expression levels were significantly lower in monocytes from MDD patients than from healthy controls ((1) vs. (3), Table 3).

The effects of antidepressant treatment on expression of microRNA regulators of TLR4 signaling were also investigated in monocytes. Similar to findings in PBMCs, let-7e, miR-146a, and miR-155 levels in monocytes were increased after antidepressant treatment ((1) vs. (2), Table 3). In addition, miR-145 level was increased, whereas miR-21-5p level was decreased, in monocytes.

### 3.4. Association between microRNA Expressions and Clinical Findings

To further investigate the association between changes of microRNA and responses to treatment, patients were divided into remission and non-remission groups. There was no difference at baseline between these two groups ((1) vs. (2), Table 4). Those who achieved remission after treatment showed significantly increased levels of let-7e, miR-223, miR-145, and miR-155 in PBMCs ((1) vs. (3), Table 4). However, the non-remission group did not show any change in these microRNAs ((2) vs. (3), Table 4).

Among the 69 patients who received four-week antidepressant treatment, 21 patients received selective serotonin reuptake inhibitors (SSRIs), whereas 32 patients received serotonin–norepinephrine reuptake inhibitors (SNRIs). SSRI treatment significantly increased levels of let-7e and miR-155 in PBMCs, whereas the SNRI treatment group had no effect. SSRIs also increased levels of miR-223 and miR-145 expression (Appendix A, Appendix A).

To explore the association between severity of depression and the expression of negative regulatory microRNAs, multiple linear regression analysis was used, and the results showed that let-7e and miR-146a were negatively correlated with HAMD-17 score, whereas miR-155 was positively correlated with HAMD-17 score (Table 5).

## 4. Discussion

In this study, we assessed the associations between MDD and the expression of intracellular microRNAs regulating TLR4 signaling in PBMCs and monocytes. We found that, prior to antidepressant treatment, only the levels of let-7e, miR-146a, and miR-155 in PBMCs were significantly lower in MDD patients than in healthy controls, and they were increased significantly after antidepressant treatment. In monocytes, only miR-146a and miR-155 showed a similar pattern. Patients who achieved remission demonstrated significantly increased let-7e and miR-155 levels in PBMCs, whereas unremitted patients showed no changes. In addition, let-7e and miR-146a levels were negatively associated with HAMD-17 scores, but miR-155 was positive correlated with severity of depression. To our knowledge, this study is the first to show that the intracellular negative regulatory microRNAs for TLR4 signaling are aberrantly expressed in patients with MDD and show simultaneous downregulation of microRNAs during acute depression and their increase after antidepressant treatment.

Our findings are similar to a postmortem brain study showing that miR-146a was downregulated in depressed subjects who committed suicide [42]. The miR-146a–TRAF6 regulatory axis is regarded as an essential molecular brake, preventing immune overreaction by attenuating NF-κB signaling [43]. Low miR-146a expression would induce excessive production of pro-inflammatory cytokines, including TNF-α and IL-6 [44], and reduce 5-hydroxytryptamine (5-HT) levels through activation of the tryptophan-metabolizing enzyme, indoleamine 2,3-dioxygenase (IDO) [45]. These results reflect the imbalance in the TLR-mediated inflammatory pathway in patients with MDD.

Regarding to the roles of let-7e in a genetic rat model of depression, a recent work reported that Flinders Sensitive Line (FSL) showed elevation of interleukin-6 and decrease of the let-7 family in the prefrontal cortex (PFC), suggesting that disturbance of the let-7 family biogenesis may function to increase proinflammatory markers in the depression [46]. In a learned helplessness rat model of depression, downregulation of let-7e within the frontal cortex was also found [47]. Moreover, let-7e was found to involve antidepressant treatment. A whole-miRNome quantitative analysis reported that let-7e was upregulated after 12 weeks of treatment with escitalopram [48]. Our data in this study concur with these previous works and prove the clinical evidence to show that microRNAs changes, such as let-7e in depressed patients during antidepressant treatment, may play an important role in the pathogenesis of psychiatric disorders [49].

While a previous study reported that loss of microRNA-155 function could reduce anxiety- and depressive-like behaviors in mice [50] and the level of miR-155 in whole blood was upregulated in individuals with depression compared with those in healthy controls, our current finding of low expression of miR-155 in PBMCs and monocytes isolated from MDD patients is in contrast to these studies. In acute central nervous system (CNS) inflammation, miR-155 upregulation was reported to be highly associated with acute microglia-mediated inflammatory responses [51]. In chronic inflammation, coordinated downregulation of the miR-155 and miR-146a genes was observed when macrophages were tolerant to endotoxin [52]. Since MDD is known to associate with chronic activation of inflammatory responses during chronic stress [4], our finding is more similar to the latter condition (Table 2). Taken together, convergence between miR-146a and miR-155 levels may be regarded as more detailed and important information for depression. However, further evidence, especially the regulation of the miR-146a and miR-155 gene promoters, is needed to explore this concept.

Clinically, we observed that HAMD-17 depression scores were negatively associated with let-7e and miR-146a levels in monocytes, but positively correlated with miR-155 levels in these cells. MiR-155 has both positive regulatory effects on inflammation, by inhibiting suppressor of cytokine signaling 1 (SOCS1), and negative regulatory effects on inflammation, by targeting myeloid differentiation primary response 88 (MyD88), TGF-β-activated kinase 1 binding protein 2 (TAB2), IκB kinase ε (IKKε), Receptor-interacting protein (RIP1), CCAAT/enhancer-binding protein beta (C/EBPβ), and SHIP1 [33]. However, miR-155 knockout mice showed reduced float duration and increased latency to float, findings reflecting less severe depression [50]. The actual role of miR-155 in MDD remains unclear, but our work has similar findings to the animal model. The present study is the first to report an association between the clinical severity of MDD and the expression of let-7e, miR-146a, and miR-155, especially in monocytes. When patients were divided into those who did and did not achieve remission after antidepressant treatment, we found that the levels of let-7e, miR-223, miR-145, and miR-155 increased significantly in those who achieved remission. These findings suggest that these microRNAs may act as biomarkers for remission during the treatment of patients with MDD. However, further prospective studies are required to test this concept.

While we were able to control for extra stimuli, including diet, smoking, and the sleep–wake cycle, by hospitalizing all patients, this study had several limitations. The sample size of MDD patients who completed the four-week treatment was not large enough, and our data only focused on the intracellular microRNA. Some studies showed that microRNA can be transferred between cells, and regulate target gene expression in the recipient cells [53]. Our data only revealed the accumulative microRNA in the cell. The expression levels of microRNAs do not represent their real function. Although we found high IL-6 levels in MDD patients, that finding could be the accumulative effects from increased TLR ligands and a decrease in other negative regulators and negative regulatory microRNAs. In addition, antidepressants and duration of MDD varied among these patients, which may have altered our results. Chronic and recurrent MDD are highly comorbid with other chronic medical illness, and the inflammation might contribute to the persistence of MDD [54]. Duration of MDD may affect inflammatory biomarker profiles. For example, elevated CRP and cytokines including IL-1β, TNF-α, and IL-6 were reported in patients with MDD. However, IL-6 [55] and CRP [56] were not elevated in drug-naïve patients with MDD. Further studies that include a larger sample size with different category groups of doses and types of antidepressants, with a focus on innate immune regulation, are warranted.

## 5. Conclusions

In summary, the present study proved once more the concept of disturbance in negative regulation for TLR4 signaling in MDD. The therapeutic implications of our findings are considerable. In conjunction with current therapeutic regimens, modulating negative regulatory microRNA expression to rebalance TLR-mediated inflammatory signaling may provide a potential approach for MDD management.

## Figures and Tables

**Table 1 cells-08-00647-t001:** Demographic and clinical characteristics of patients with major depressive disorder and healthy controls. MDD—major depressive disorder; M—male; F—female; BMI—body mass index; HAMD-17—17-item Hamilton Depression Rating Scale.

	(1) MDD before Treatment (*n* = 84)	(2) MDD after Treatment (*n* = 69)	(3) Healthy Controls (*n* = 43)	*p*-Value		(4) MDD Remission (*n* = 31)	(5) MDD Non-Remission (*n* = 38)	*p*-Value
				(1) vs. (3)	(2) vs. (3)			(4) vs. (5)
Age (years)	45.20 ± 11.00	45.56 ± 10.46	41.88 ± 9.03	*p* = 0.086	*p* = 0.059	45.39 ± 9.85	45.71 ± 11.06	*p* = 0.899
Sex (M/F)	20/64	16/63	6/37	*p* = 0.251	*p* = 0.793	8/23	4/34	*p* = 0.098
BMI (kg/m^2^)	24.61 ± 4.26	24.82 ± 4.19	23.91 ± 3.25	*p* = 0.259	*p* = 0.195	24.55 ± 4.17	25.05 ± 4.23	*p* = 0.624
Smoking (yes/no)	30/54	22/47	5/38	*p* = 0.005 *	*p* = 0.015 *	13/18	9/29	*p* = 0.087
HAMD-17	24.16 ± 5.48	8.91 ± 5.08	-	-	-	22.90 ± 5.17	25.45 ± 5.69	*p* = 0.056

Results are reported as means ± SD or as numbers. Age and BMI were compared by Student’s *t*-tests. Sex and smoking were compared by the chi-squared test; * *p* < 0.05.

**Table 2 cells-08-00647-t002:** Expression in peripheral blood mononuclear cells (PBMCs) of individual microRNAs regulating Toll-like receptor (TLR) signaling in patients with major depressive disorder and healthy controls. IL-6—interleukin-6.

	(1) MDD before Treatment (*n* = 84)	(2) MDD after Treatment (*n* = 69)	(3) Healthy Controls (*n* = 43)	(1) vs. (3)	(1) vs. (2)
				F- and *p*- Values	*p*-Value
let-7e	−4.09 ± 1.76	−3.62 ± 1.40	−3.41 ± 1.16	F = 4.605 *p* = 0.034 *	*p* = 0.002 *
miR-21-5p	−6.00 ± 2.11	−5.54 ± 2.12	−5.28 ± 1.18	F = 4.097 *p* = 0.045 *	*p* = 0.062
miR-223	2.95 ± 1.56	3.36 ± 1.49	3.41 ± 0.77	F = 2.906 *p* = 0.091	*p* = 0.002 *
miR-145	−5.75 ± 1.43	−5.61 ± 1.50	−5.17 ± 1.25	F = 3.748 *p* = 0.055	*p* = 0.111
miR-146a	−1.88 ± 2.06	−1.63 ± 2.06	−0.60 ± 0.85	F = 15.374 *p* = 0.000 *	*p* = 0.038 *
miR-155	−3.23 ± 1.78	−2.90 ± 1.43	−2.25 ± 0.77	F = 11.386 *p* = 0.001 *	*p* = 0.004 *
IL-6	−9.50 ± 1.80	−9.73 ± 1.70	−10.13 ± 1.48	F = 5.113 *p* = 0.026 *	*p* = 0.025 *

Results are reported as means ± SD; (1) vs. (3) was compared by analysis of covariance (ANCOVA) after adjustment for age, sex, smoking, and BMI; F-value is defined as between-group variance divided by within-group variance; (1) vs. (2) was compared by paired *t*-tests; * *p* < 0.05.
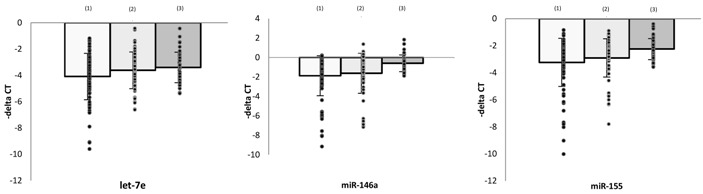

**Table 3 cells-08-00647-t003:** Expression in monocytes of individual microRNAs regulating TLR signaling in patients with major depressive disorder and healthy controls.

	**(1) MDD before Treatment (*n* = 47)**	**(2) MDD after Treatment (*n* = 33)**	**(3) Healthy Controls (*n* = 33)**	**(1) vs. (3)**	**(1) vs. (2)**
				**F- and *p−* Values**	***p*−Value**
let−7e	−2.52 ± 0.79	−1.70 ± 0.82	−2.10 ± 0.92	F = 3.088 *p* = 0.083	*p* = 0.001 *
miR−21−5p	−3.78 ± 1.38	−5.16 ± 1.22	−3.53 ± 0.75	F = 0.394 *p* = 0.532	*p* = 0.022 *
miR−223	5.15 ± 0.77	5.44 ± 0.78	5.14 ± 0.80	F = 0.128 *p* = 0.722	*p* = 0.054
miR−145	−6.17 ± 1.42	−5.44 ± 0.97	−5.58 ± 0.93	F = 2.932 *p* = 0.091	*p* = 0.006 *
miR−146a	−2.71 ± 1.55	−2.27 ± 0.95	−1.48 ± 1.23	F = 12.320 *p* = 0.001 *	*p* = 0.034 *
miR−155	−2.57 ± 1.17	−2.11 ± 0.685	−1.72 ± 0.95	F = 10.208 *p* = 0.002 *	*p* = 0.025 *

Results are reported as means ± SD; (1) vs. (3) was compared by ANCOVA after adjustment for age, sex, smoking, and BMI; F-value is defined as between-group variance divided by within-group variance; (1) vs. (2) was compared by paired *t*-test; * *p* < 0.05.
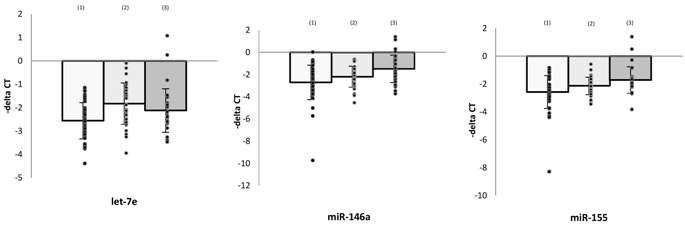

**Table 4 cells-08-00647-t004:** Expression in PMBCs of microRNAs negatively regulating TLR signaling before and after treatment with antidepressants in patients experiencing remission (*n* = 31) and non-remission (*n* = 38).

	Before Antidepressant Treatment		After Antidepressant Treatment			*p*−Value		
	(1) Remission	(2) Non−Remission	(3) Remission		(4) Non−Remission	(1) vs. (2)	(1) vs. (3)	(2) vs. (4)
let−7e	−4.38 ± 1.80	−4.16 ± 1.80	−3.56 ± 1.21	−3.70 ± 1.54	F = 0.332 *p* = 0.567	*p* = 0.002 *	*p* = 0.115	
miR−21−5p	−6.16 ± 2.02	−6.03 ± 2.20	−5.44 ± 2.42	−5.62 ± 1.87	F = 0.040 *p* = 0.842	*p* = 0.135	*p* = 0.296	
miR−223	3.00 ± 1.69	2.77 ± 1.72	3.64 ± 1.64	3.10 ± 1.34	F = 0.197 *p* = 0.659	*p* = 0.001 *	*p* = 0.177	
miR−145	−6.05 ± 1.40	−5.82 ± 1.46	−5.44 ± 1.42	−5.75 ± 1.58	F = 0.252 *p* = 0.618	*p* = 0.020 *	*p* = 0.820	
miR−146a	−1.90 ± 2.27	−2.05 ± 2.16	−0.92 ± 1.00	−1.73 ± 2.19	F = 0.036 *p* = 0.849	*p* = 0.107	*p* = 0.202	
miR−155	−3.32 ± 1.58	−3.38 ± 2.11	−2.85 ± 1.31	−2.98 ± 1.55	F = 0.000 *p* = 0.992	*p* = 0.001 *	*p* = 0.115	

Results are reported as means ± SD; (1) vs. (2) was compared by ANCOVA after adjustment for age, sex, smoking, and BMI; F-value is defined as between-group variance divided by within-group variance; (1) vs. (3) and (2) vs. (4) were compared by paired *t*-tests; * *p* < 0.05.
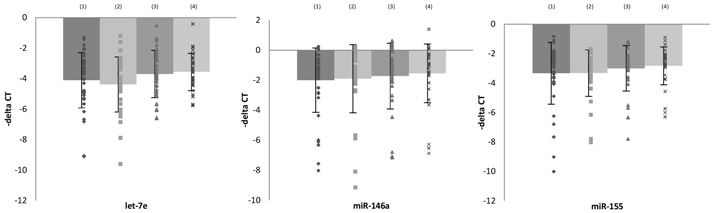

**Table 5 cells-08-00647-t005:** Correlations between HAMD-17 scores and expression of microRNA in monocytes as determined by multiple linear regression analysis.

Independent Factors	HAMD-17 Score		
	Standardized Coefficients	*t*	*p*−Value
let−7e	−0.793	−2.946	*p* = 0.006 *
miR−21−5p	0.004	0.012	*p* = 0.990
miR−223	0.316	0.793	*p* = 0.434
miR−145	−0.027	−0.114	*p* = 0.910
miR−146a	−1.111	−3.500	*p* = 0.002 *
miR−155	1.001	2.886	*p* = 0.007 *

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
