# Peer review of "Aberrant Expression of Intracellular let-7e, miR-146a, and miR-155 Correlates with Severity of Depression in Patients with Major Depressive Disorder and Is Ameliorated after Antidepressant Treatment"

_cells, 2019, doi:10.3390/cells8070647_

Round 1

Reviewer 1 Report

The authors address a very interesting issue that is the role played by miRNA in worsening MDD symptoms. 

Results are interesting and novel, and methods are scientifically sound. 

However, the manuscript may be improved by up-dating reference List, that must include papers published in 2018 and 2019, and adding in the Introducton and Discussion several citations of papers reporting on the role of different miRNAs in the response to antidepressant treatment. 

Author Response

The manuscript may be improved by up-dating reference List, that must include papers published in 2018 and 2019, and adding in the Introduction and Discussion several citations of papers reporting on the role of different miRNAs in the response to antidepressant treatment. 

Thank you for your suggestion. We have added and updated the references (22, 25, 26, 27, 30, 31, 32, and 33) in the sections of Introduction and Discussion with papers published in 2018 and 2019 to elucidate the role of miRNAs in the response to antidepressant treatment as reviewer’s suggested.

Reviewer 2 Report

The main problem with this paper is the presentation of the results. It is basically impossible to judge the differences between the various groups in terms of miRNA levels based on the numbers presented in table format. To better illustrate the differences, a dot plot should be presented instead of (or in addition to) the tables.

What is F? This is never defined--I suppose it is some type of fold change but it is very difficult to tell how it is calculated. For example, in Table 2, let-7e levels in group 1 vs group 3 are -4.09+/-1.76 vs -3.41+/-1.16.  How does one get an F value of 4.605? 

Author Response

Reviewer 2

To better illustrate the differences, a dot plot should be presented instead of (or in addition to) the tables.

Thanks for reviewer’s suggestion. In addition to the tables, we have added the dot plots as the attached file for the reviewer.

What is F? This is never defined--I suppose it is some type of fold change but it is very difficult to tell how it is calculated. For example, in Table 2, let-7e levels in group 1 vs group 3 are -4.09+/-1.76 vs -3.41+/-1.16.  How does one get an F value of 4.605?

Thank you for your comments. We used the SPSS software to calculate our statistical analysis. In ANCOVA test, F = Test statistic = MSfixed factor/MSerror

Here we included the original data of let-7 for your reference as the following attached table.

Reviewer 3 Report

Chronic inflammation and abnormalities in Toll-like receptor (TLR) signaling pathways have been associated with major depressive disorder (MDD). Previous studies reported that impaired negative regulators for the TLR pathways are associated with MDD. In the present study Authors aimed to assess the association between the severity of depression and the intracellular microRNAs that regulate TLR4 signaling in both peripheral blood mononuclear cells (PBMCs) and monocytes from MDD patients.

Overall, I found the present study timely, interesting and scientifically sound. However, I have some comments that aim to improve the high quality of the study:

1) Recently, a possible relationship between C-Reactive Protein (CRP), a marker of underlying low-grade inflammation, and mood disorders has been proposed and widely studied. I suggest Authors to add a brief comment on this with appropriate references (see De Berardis et al.Int J Immunopathol Pharmacol. 2006 Oct-Dec;19(4):721-5 and De Berardis et al. CNS Spectr. 2017 Aug;22(4):342-347).

2) The sentence "Remission was defined as total HAMD-17 score ≤ 7" needs a reference.

3) How many patients were screened, but refused to participate or excluded? I suggest Authors to add more information on how patients were included.

4) There are no specified inclusion/exclusion criteria. Please add this part.

5) Was mental retardation assessed and considered as an exclusion criterion? Please specify.

6) Concerning demographic, the duration of MDD would be an important variable for the study that should be discussed and evaluated as this may influence results.

Author Response

1) Recently, a possible relationship between C-Reactive Protein (CRP), a marker of underlying low-grade inflammation, and mood disorders has been proposed and widely studied. I suggest Authors to add a brief comment on this with appropriate references (see De Berardis et al.Int J Immunopathol Pharmacol. 2006 Oct-Dec;19(4):721-5 and De Berardis et al. CNS Spectr. 2017 Aug;22(4):342-347).

Thank you for your suggestion. We have described C-Reactive Protein (CRP) in the mood disorders and updated the reference in the section of Introduction (reference 7 and 8) as the reviewer suggested.

2) The sentence "Remission was defined as total HAMD-17 score ≤ 7" needs a reference.

Thank you for your suggestion. We have added the reference (reference 42) as reviewers suggested.

3, 4) How many patients were screened, but refused to participate or excluded? I suggest Authors to add more information on how patients were included. There are no specified inclusion/exclusion criteria. Please add this part.

Thank you for your suggestion. A total of 84 patients with MDD were enrolled, including 20 men and 64 women. Of these, 69 patients were treated with an antidepressant for 4 weeks and returned for follow-up examination. We have added the following sentences in our Materials and Methods section as reviewers suggested. “Patients with MDD were screened and enrolled by two psychiatrists before entering the study. Briefly, screening included a Structured Clinical Interview for DSM-V Axis I Disorders, an assessment of current psychiatric symptoms, and a determination of previous antidepressant treatment. Patients with psychotic disorder, mental retardation, substance dependence, severe metabolic syndrome, a body mass index (BMI)>34 kg/m2, a history of any systemic inflammatory disease, or who were taking anti-inflammatory or immune modulating drugs were excluded from the study.”

5) Was mental retardation assessed and considered as an exclusion criterion? Please specify.

Mental retardation was assessed and considered as an exclusion criterion.

6) Concerning demographic, the duration of MDD would be an important variable for the study that should be discussed and evaluated as this may influence results.

We fully agreed with the reviewer’s comments. In the discussion section, we have discussed the duration of MDD as an important variable for our study since it may influence results as described in page 8 line 324-328: “Chronic and recurrent MDD are highly comorbid with other chronic medical illness and the inflammation might contribute to the persistence of MDD [55]. Duration of MDD may affect inflammatory biomarkers profiles. For example, elevated CRP and cytokines including IL-1β TNF-α and IL-6 were reported in patients with MDD. However, IL-6 [56] and CRP [57] were not elevated in drug-naïve patients with MDD.”

Round 2

Reviewer 2 Report

The authors only provide dot plots to represent the data from Table 2.  This should be done for the data in Tables 3 and 4 as well.

Regarding the definition of "F", simply stating "we ran this statistical test and the value we got was F..." is not sufficient.  Please talk about what F means in the Statistical Analysis portion of your Methods section.  A scientist with a moderate level of understanding of statistics should be able to follow what was done and what the values represent.

Author Response

Point 1: The authors only provide dot plots to represent the data from Table 2.  This should be done for the data in Tables 3 and 4 as well. Regarding the definition of "F", simply stating "we ran this statistical test and the value we got was F..." is not sufficient.  Please talk about what F means in the Statistical Analysis portion of your Methods section.  

Response 1: Thanks for your suggestion. 

1. We have added dot plots to table 3 and Table 4 as reviewer suggested.

2. We have added the definition of F value in the section of Materials and Methods, and described it in the results and tables that help the readers to understand:  

"The F value is defined as between-groups variance divided by within-groups variance."

Reviewer 3 Report

The paper is much improved and worthy of publication in Cells

Author Response

Thank you for your valuable suggestion. Your comments and approval is highly appreciated.